# Genome-wide study of a Neolithic Wartberg grave community reveals distinct HLA variation and hunter-gatherer ancestry

Alexander Immel[1], Federica Pierini[2], Christoph Rinne[3], John Meadows [4,5], Rodrigo Barquera [6], András Szolek [7], Julian Susat[1], Lisa Böhme[1], Janina Dose[1], Joanna Bonczarowska[1], Clara Drummer[3], Katharina Fuchs[1], David Ellinghaus [1], Jan Christian Kässens[1], Martin Furholt[8], Oliver Kohlbacher [7,9,10,11], Sabine Schade-Lindig[12], Andre Franke[1], Stefan Schreiber[1,13], Johannes Krause [6], Johannes Müller[3], Tobias L. Lenz [2], Almut Nebel[1] & Ben Krause-Kyora [1✉]

The Wartberg culture (WBC, 3500-2800 BCE) dates to the Late Neolithic period, a time of important demographic and cultural transformations in western Europe. We performed genome-wide analyses of 42 individuals who were interred in a WBC collective burial in Niedertiefenbach, Germany (3300-3200 cal. BCE). The results showed that the farming population of Niedertiefenbach carried a surprisingly large hunter-gatherer ancestry component (34–58%). This component was most likely introduced during the cultural transformation that led to the WBC. In addition, the Niedertiefenbach individuals exhibited a distinct human leukocyte antigen gene pool, possibly reflecting an immune response that was geared towards detecting viral infections.

[1] Institute of Clinical Molecular Biology, Kiel University, Rosalind-Franklin-Strasse 12, 24105 Kiel, Germany. [2] Research Group for Evolutionary Immunogenomics, Max Planck Institute for Evolutionary Biology, August-Thienemann-Strasse 2, 24306 Plön, Germany. [3] Institute of Pre- and Protohistoric Archaeology, Kiel University, Johanna-Mestorf-Strasse 2-6, 24118 Kiel, Germany. [4] Leibniz Laboratory for AMS Dating and Isotope Research, Kiel University, Max-Eyth-Strasse 11-13, 24118 Kiel, Germany. [5] Centre for Baltic and Scandinavian Archaeology (ZBSA), Schloss Gottorf, 24837 Schleswig, Germany. [6] Max Planck Institute for the Science of Human History, Khalaische Strasse 10, 07745 Jena, Germany. [7] Applied Bioinformatics, Department for Computer Science, University of Tübingen, Sand 14, 72076 Tübingen, Germany. [8] Department of Archaeology, Conservation and History, University of Oslo, Blindernveien 11, 0371 Oslo, Norway. [9] Institute for Bioinformatics and Medical Informatics, University of Tübingen, Sand 14, 72076 Tübingen, Germany. [10] Institute for Translational Bioinformatics, University Hospital Tübingen, Hoppe-Seyler-Strasse 9, 72076 Tübingen, Germany. [11] Biomolecular Interactions, Max Planck Institute for Developmental Biology, Max-Planck-Ring 5, 72076 Tübingen, Germany. [12] Landesamt für Denkmalpflege Hessen, hessenARCHÄOLOGIE, Schloss Biebrich, 65203 Wiesbaden, Germany. [13] Department of General Internal Medicine, University Hospital Schleswig-Holstein, Kiel University, Rosalind-Franklin-Strasse 12, 24105 Kiel, Germany. ✉email: b.krause-kyora@ikmb.uni-kiel.de

Over the last few years, large-scale ancient DNA (aDNA) studies have provided unprecedented insights into the peopling of Europe and the complex genetic history of its past and present-day inhabitants[1–5]. Recent research has particularly focused on population dynamics during the Neolithic period. The first agriculturalists, who appeared with the uniform Linear Pottery culture (Linearbandkeramik, LBK, 5450–4900 BCE) across central Europe, probably co-existed with local hunter–gatherers (HGs) for about 2000 years[6]. Although both groups are thought to have lived in close proximity, initially only limited admixture occurred[2,3]. This situation changed later (4400–2800 BCE) when the gene pool of the early farmers was transformed through the introgression of genomic components typical of HG populations[1,3,7].

The Late Neolithic period is archaeologically characterized by strong regional diversification and a patchwork of small units of classification (i.e., archaeological cultures)[8]. One of the western units that emerged at the beginning of the Late Neolithic period is associated with the Wartberg culture (WBC, 3500–2800 BCE), which most likely developed from the Late Michelsberg culture (MC, 3800–3500 BCE)[9,10]. WBC is mainly found in western central Germany (Fig. 1)[11,12]. It is known for its megalithic architecture of large gallery graves that is distinct from that in adjacent regions, but shows a striking resemblance to similar monuments in the Paris Basin and Brittany[13,14]. Despite the central geographical location of WBC that connects cultural influences from several directions, no genome-wide data of human remains from WBC sites have so far been investigated.

We performed genome-wide analyses of 42 individuals who were buried in a WBC gallery grave near the township of Niedertiefenbach in Hesse, Germany (Fig. 1), dated between 3300 and 3200 cal. BCE (Supplementary Information). In contrast to other genome-wide aDNA studies, which usually include a small number of individuals from a specific site and period, we provide a snapshot of a burial community that used the collective grave for ~100 years[15] (Supplementary Information, Supplementary Fig. 1 and 2, and Supplementary Table 1). In addition to performing population genetic and kinship analyses, we also investigated the immune-relevant human leukocyte antigen (HLA) region. This approach allowed us not only to reconstruct the genetic ancestry of the WBC-associated people from Niedertiefenbach but also to gain insights into the makeup of immunity-related genes of a Late Neolithic group.

## Results

In the framework of this study, we dated 25 individuals from the Niedertiefenbach collective. A Bayesian modeling suggested that all fell into the range 3300–3200 cal. BCE[15] (Supplementary Information, Supplementary Table 1, and Supplementary Fig. 2). aDNA extracts obtained from 89 randomly selected individuals[16,17] were subjected to shotgun sequencing. Of these, we filtered out 47 who (i) had fewer than 10,000 single-nucleotide polymorphisms (SNPs) benchmarked on a previously published dataset of 1,233,013 SNPs[1,2,4] or (ii) showed evidence of X-chromosomal contamination and/or contamination on the mitochondrial level (≥5%). Thus, after quality control, genomic data from 42 individuals were used for subsequent analyses (Supplementary Data 1). DNA damage patterns were consistent with an ancient origin[18] of the isolated DNA fragments. As part of our standardized analysis pipeline, we also screened the sequence data for known blood-borne pathogens. No molecular

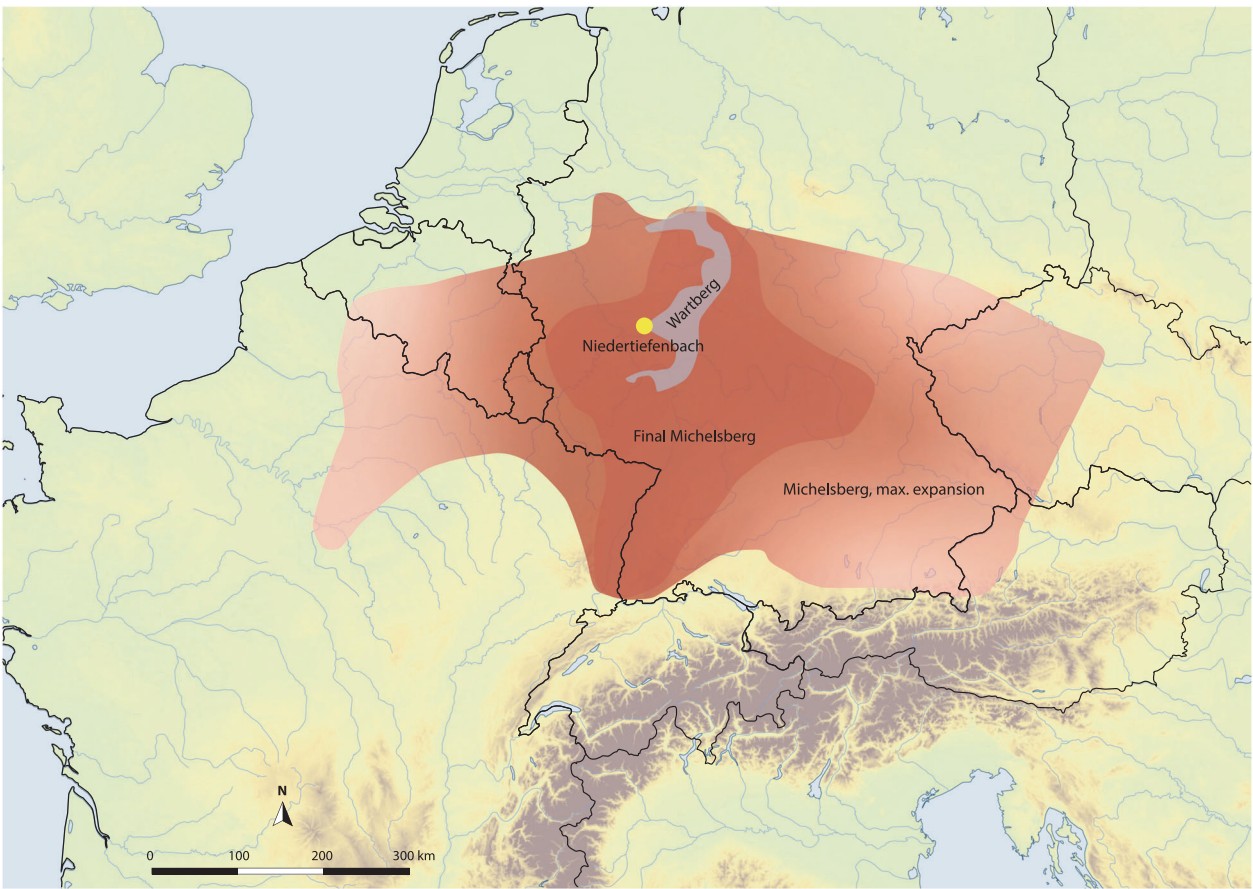

**Fig. 1 Map with the site Niedertiefenbach.** Geographic distribution of the archaeological small units mentioned in this study are shown.

evidence for pathogens was detected. Ten individuals were genetically determined as females and 25 as males. In seven cases, the genetic sex could not be clarified due to missing or low sequence coverage of the sex chromosomes. Therefore, we integrated osteological information on skeletal sex that resulted in the identification of one additional female and one additional male. Thus, in the whole set of 42 individuals, 11 were determined as females and 26 as males (Supplementary Data 1).

First, the SNP information derived from the Niedertiefenbach collective together with previously published datasets of 122 ancient populations was projected onto a basemap calculated from 59 modern-day West-Eurasian populations applying principal component analysis (PCA)[1,2,4,19]. The Niedertiefenbach individuals formed a cluster that is mainly explained by genetic variation between HGs and early farmers on the first principal component (Fig. 2). However, the Niedertiefenbach samples covered a wide genetic space which reflects a high intra-population diversity. Some of the individuals grouped closely with two specimens from Blätterhöhle, a cave site near Hagen, Germany (4100–3000 BCE)[6] (Fig. 2). ADMIXTURE analysis[20] with four to eight components suggested two main genetic contributions to the Niedertiefenbach collective—one maximized in European HGs and the other in Neolithic farmers from Anatolia (Supplementary Fig. 3). Next we applied $f3$ outgroup statistics[21] to calculate the amount of shared genetic drift between the Niedertiefenbach individuals and another test population relative to an outgroup [$f_3$(Niedertiefenbach; test; Mbuti)]. The highest amount of shared genetic drift was observed between Niedertiefenbach and European HGs from Sicily, Croatia, and Hungary (Supplementary Fig. 4). To estimate the amount of Neolithic farmer and HG genetic ancestry in the Niedertiefenbach group, we ran qpAdm[21]. We obtained feasible models for Niedertiefenbach as a two-way mixture of Neolithic farmers from Anatolia and various European HGs (P range between 0.1 and 0.74), which altogether provided on average ~60% Neolithic Anatolian farmer ancestry and ~40% HG ancestry (Supplementary Data 2). Another feasible two-way admixture model for Niedertiefenbach was the combination of Anatolian Neolithic farmers (41%) and individuals from Blätterhöhle (P = 0.17). We then applied ALDER[22] to estimate the date of the mixture of ancestry components associated with early farmers and the Loschbour HG (Waldbillig, Luxembourg) as a representative of western HGs in the Niedertiefenbach population. ALDER suggested 14.85 ± 2.82 generations (Supplementary Information) before the ¹⁴C benchmark of 3300–3200 cal. BCE. We confirmed this result using the software DATES[23] which yielded a similar number of generations (16.6 ± 2.65) (Supplementary Information). Based on a generation time of 29 years[24], the date for the emergence of the genetic composition of the Niedertiefenbach community appeared to be between 3860 and 3550 cal. BCE.

For phenotype reconstruction, we investigated selected SNPs associated with skin pigmentation and hair color (rs16891982), eye color (rs12913832), starch digestion (rs11185098), and lactase persistence (rs4988235)[25]. Not all of these SNPs were available for all of the investigated individuals due to poor sequence coverage. Fourteen of the 42 individuals carried only the rs16891982-C allele, which is associated with dark hair and increased skin pigmentation[26], whereas three had both alleles (C and G). Only three individuals carried the rs12913832-G allele associated with blue eye color, seven had the A allele associated with brown eye color, and eight had both alleles. The minor A allele of rs11185098 is positively associated with AMY1 (amylase 1) gene copies and high amylase activity responsible for starch digestion[27]. Only one individual was found to be homozygous for the G allele and six had both alleles, whereas no homozygous carrier for the A allele was found. All individuals with enough coverage

for rs4988235 carried the G allele that tags an ancestral haplotype associated with lactase non-persistence[28], which suggests that the Niedertiefenbach people could not digest dairy products.

To determine the HLA class I and II alleles of the Niedertiefenbach individuals, we applied a previously developed method[29]. In addition, we used OptiType, an automated HLA-typing tool[30]. Only alleles that were consistently called by both methods were considered for the analysis. We successfully genotyped alleles at the three classical HLA class I loci A, B, and C, and the three class II loci DPB1, DQB1, and DRB1 in 23 unrelated individuals (Supplementary Data 3). For each of the six HLA loci, the two most common alleles differed in frequency to the modern-day German population by at least 9% (Table 1), suggesting substantial differences between the ancient and modern HLA allele pools, even when considering the 95% confidence intervals (CIs). Using proxy SNPs, it was possible for 7 of the 12 alleles to trace their frequency over time in published datasets[25,31]. We observed that five of them, i.e., alleles at HLA-B, -C, and -DRB1, appeared at much higher frequencies in HGs (≥47%) than in the Niedertiefenbach samples or in early farmers (Fig. 3A and Supplementary Data 4). In line with this finding, these alleles were even less frequent in present-day Germans (Table 1), many of them statistically significantly so, such as HLA-C*01:02. Interestingly, this allele also showed some of the highest phylogenetic divergence among the HLA-C alleles observed in the Niedertiefenbach individuals (Fig. 3B). Elevated sequence divergence between the amino acid sequences of the two alleles at a given HLA locus is a proxy for larger functional differences between the encoded HLA molecule variants, leading to a larger overall range of presented antigens, and has been associated with higher immunocompetence[32,33]. Indeed, Niedertiefenbach individuals whose HLA-C genotype included the allele C*01:02 exhibited a higher divergence between their HLA-C alleles than those who did not carry this allele (Mann-Whitney U test, P = 0.018; Fig. 3C). Furthermore, the HLA alleles in the Niedertiefenbach individuals appeared to bind a particular unique set of viral peptides (Supplementary Fig. 5). Similar patterns were observed also for the most frequent allele at the HLA-B locus, B*27:05 (Supplementary Fig. 6). Overall, the frequency differences for these two and other alleles led to a significant difference in allele pool composition between the Niedertiefenbach individuals and modern Germans (analysis of similarity, ANOSIM, P = 0.001; Supplementary Fig. 7).

We noted 29 different mitochondrial DNA (mtDNA) haplogroups and five Y chromosome haplotypes, all of which belonged to haplogroup I2 (Supplementary Data 1). Ten of the 16 males for whom high-resolution Y haplotype information could be generated carried the same haplotype (I2c1a1). We performed kinship analyses using f3 statistics[21] and READ[34]. Both programs identified one triplet consisting of a female and two males as first-degree relatives (Supplementary Fig. 8). Parent–child relationships could be ruled out as the osteological analysis showed that all three individuals had died in infancy (age at death 1–3 years) or early childhood (4–6 years). This left the sibling constellation as the only other possible explanation, which was supported by the respective mtDNA and Y chromosome haplotypes, as well as the HLA allele profiles.

## Discussion

It has clearly been established that the transformation from the LBK, which is characterized by a homogeneous material culture over a large area, to the later more diverse Neolithic societies in Europe was accompanied by genetic admixture[3]. However, the population interactions underlying this transformation have not yet been fully resolved. The admixture events were geographically

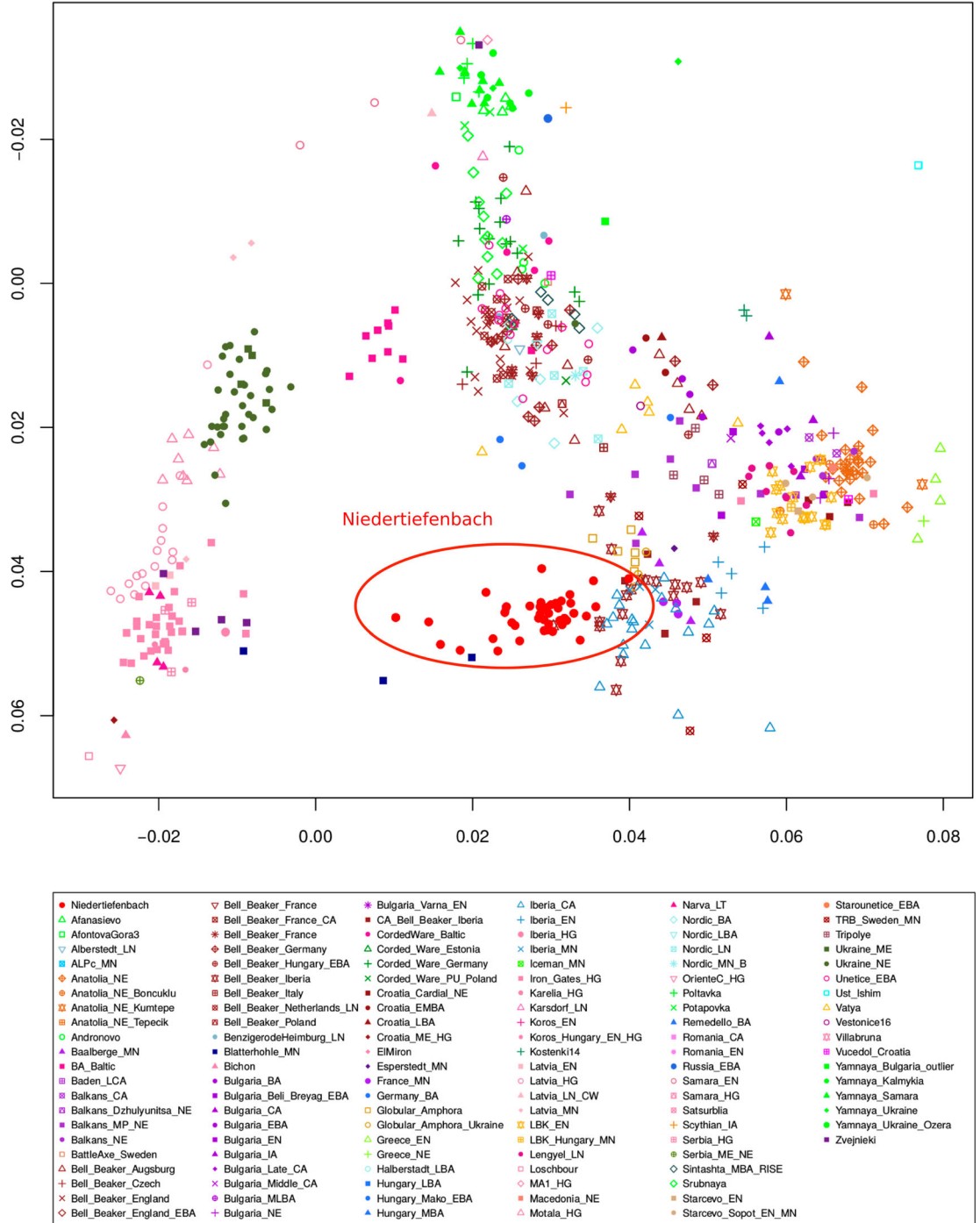

**Fig. 2 PCA of individuals from 123 ancient populations including Niedertiefenbach.** PCA of individuals were projected onto the first two principal components calculated from 59 present-day West-Eurasian populations (not shown for clarity). Niedertiefenbach individuals are depicted as red circles.

highly localized and involved various populations with different ancestry components[3]. These processes likely led to the increase in HG ancestry proportions and mtDNA lineages that were observed in Middle to Late Neolithic communities[1,7,35,36]. It is currently not known what might have influenced these widespread demographic and genomic processes in Europe, but climate change and/or social processes may be considered contributing factors[37].

Here we investigated a community of 42 Late Neolithic farmers excavated from the WBC gallery grave in Niedertiefenbach, Germany[16,17,38]. The radiocarbon dates (3300–3200 cal. BCE)

placed the site only a few hundred years before the arrival of the steppe ancestry in central Europe. Interestingly, we did not observe any genetic evidence for an admixture with steppe components (e.g., no feasible two-way models with steppe ancestry or the absence of the Y haplogroup R1b). The Niedertiefenbach population exhibited a mixture of genomic components from western HGs and early farmers. The continuous range (34–58%) of the relatively high genetic HG proportion is surprising. Admixture dating indicated that the mixing of the two components occurred around 3860–3550 cal. BCE. From these results, it cannot be inferred to what extent the contributing

**Table 1 Frequency of the two most common alleles of the six HLA loci of the Niedertiefenbach individuals compared to a modern-day German population.**

| HLA locus | Allele | Allele count in Niedertiefenbach (46 alleles) | Allele frequency Niedertiefenbach (%; N = 23) | Lower and upper limit of the 95% confidence interval | Frequency today (%; N = 3219)[a] | Lower and upper limit of the 95% confidence interval | Adj. P-value* |
|---|---|---|---|---|---|---|---|
| A | 02:01 | 29 | 63.04 | 49.08–77.00 | 29.81 | 28.69–30.93 | $2.5 \times 10^{-5}$ |
| A | 24:02 | 9 | 19.57 | 8.10–31.04 | 9.34 | 8.63–10.05 | 0.413 |
| B | 27:05 | 11 | 23.91 | 11.58–36.24 | 4.54 | 4.03–5.05 | $5.8 \times 10^{-8}$ |
| B | 51:01 | 9 | 19.57 | 8.10–31.04 | 5.28 | 4.73–5.82 | $9.4 \times 10^{-4}$ |
| C | 01:02 | 8 | 17.39 | 6.43–28.35 | 3.06 | 2.64–3.47 | $3.8 \times 10^{-6}$ |
| C | 02:02 | 8 | 17.39 | 6.43–28.35 | 5.72 | 5.15–6.28 | 0.027 |
| DPB1 | 02:01 | 17 | 36.96 | 23.00–50.92 | 12.44 | 11.64–13.24 | $2.2 \times 10^{-5}$ |
| DPB1 | 04:01 | 11 | 23.91 | 11.58–36.24 | 45.15 | 44.2–46.63 | 0.075 |
| DQB1 | 03:01 | 19 | 41.30 | 27.07–55.53 | 18.56 | 17.62–19.5 | 0.002 |
| DQB1 | 04:02 | 13 | 28.26 | 15.25–41.27 | 2.87 | 2.45–3.28 | $1.7 \times 10^{-20}$ |
| DRB1 | 08:01 | 13 | 28.26 | 15.25–41.27 | 2.52 | 2.12–2.91 | $1.0 \times 10^{-23}$ |
| DRB1 | 11:01 | 10 | 21.74 | 9.82–33.66 | 8.78 | 8.09–9.46 | 0.057 |

[a]Imputed HLA allele frequency from a modern German cohort[76].
*P-value from $\chi^2$-test corrected for testing of 12 alleles.

populations themselves were already admixed or which subsistence economy they practiced. But interestingly, the estimated admixture date coincides with farming expansion phases and social changes during the Late MC (3800–3500 BCE)[39]. Archaeologically, there is a well-documented continuity from Late MC to WBC[9]. mtDNA data from two MC sites in France[40] and Germany[41] indicate that the analyzed individuals belonged to an already admixed population comprising haplotypes typical of both farmers and HGs[40]. Human genome-wide datasets from clear archaeological MC contexts are not available yet. A possible exception could be the data of four individuals from Blätterhöhle that may be chronologically (based on their radiocarbon dates of 4100–3000 BCE) and geographically linked with Late MC and/or WBC[6]. However, it has to be kept in mind that the remains were found in a cave without any definite cultural assignment. Our analyses showed that the Niedertiefenbach population appeared most closely related to the Blätterhöhle collective whose large HG component (39–72%)[3] falls into the range observed for Niedertiefenbach. Moreover, they are a good proxy for the HG component in the Niedertiefenbach sample. In addition, our admixture date is very similar to the one obtained for Blätterhöhle that yielded 18–23 generations before the average sample date of 3414 ± 84 cal. BCE[3]. Thus, there is a possible genetic link between the people buried in Blätterhöhle and those in the gallery grave of Niedertiefenbach.

The WBC-associated population in Niedertiefenbach represents a genetically diverse group with a very broad range of HG proportions (as seen in Fig. 2, Supplementary Fig. 3, and in the qpAdm two-way models for each individual in Supplementary Fig. 9). This finding suggests that the admixture was still in progress at that time or had taken place a few generations before. This scenario is tentatively supported by the admixture dating analysis (Supplementary Information). Given the surprisingly large HG component, it seems conceivable that the admixture included also individuals who had exclusive or near-exclusive genetic HG ancestry. Taking into account all available lines of evidence, we hypothesize that the increase in the HG component likely occurred during the consolidation of the MC and/or the beginning of the WBC and could have involved also direct gene flow from unadmixed local western HGs into expanding farming populations.

The genetic data of the Niedertiefenbach sample, along with information obtained from archaeological and osteological analyses, shed light on the community that used this gallery grave. In total, the skeletal remains of a minimal number of 177 individuals were recovered from the 7 m$^2$ site, reflecting a very high occupancy rate for a collective WBC burial[38]. The genetic sex distribution in the sample indicated a considerable excess of males (70%) among adults and subadults, which has not been described for other Neolithic populations[42]. As we followed a random sampling strategy, such an excess is noteworthy and may reflect a burial bias. Regarding age, we did not observe a numerical deficit of children that is often recorded for Neolithic cemeteries in Germany[43,44]. The phenotype reconstruction revealed that the examined individuals had a predominantly dark complexion and were genetically not yet adapted to digest starch-rich food or lactose. These phenotypes have typically been described for HGs and early farmers[3].

Overall, the genomic data indicate that the gallery grave was mainly used by not closely related people who may have lived in various neighboring locations. This observation is supported by the large number of mtDNA haplogroups. However, also related individuals were interred. In one case, we observed inhumations of first-degree relatives (Supplementary Fig. 8). In addition, the presence of only one frequent Y chromosome haplotype (I2c1a1) suggests a patrilineage.

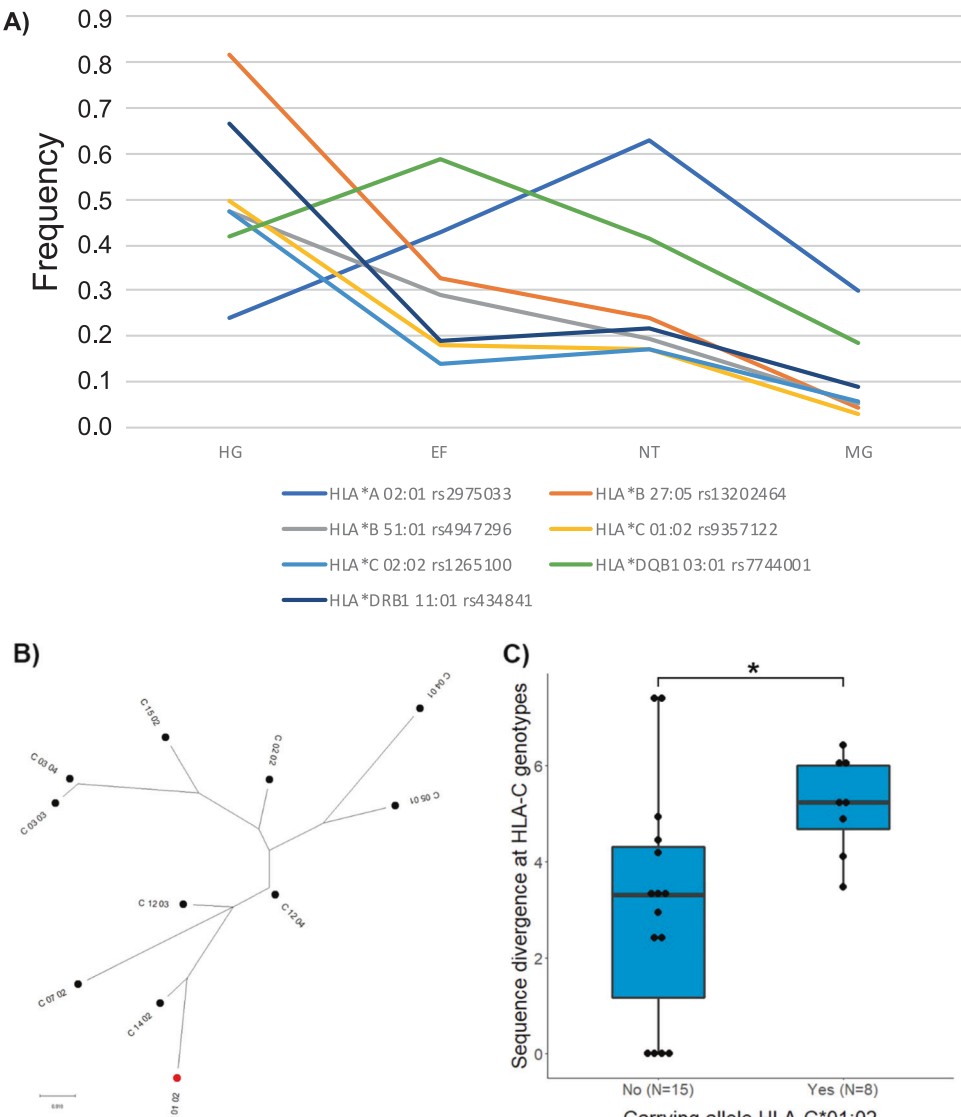

**Fig. 3 Analysis of HLA alleles. A** Frequency changes of seven HLA alleles over time. Proxy SNPs were used to determine the frequency in 140 hunter–gatherers (HG) and 455 early farmers (EF)[25]. The frequencies for the Niedertiefenbach (NT) population are based on sequencing of the HLA region. HLA frequencies of modern German individuals (MG) were imputed using standard methods[76] (Supplementary Data 4). **B** Phylogenetic relationship of the 11 HLA-C alleles observed among the Niedertiefenbach individuals. Maximum-likelihood tree based on amino acid sequence (HLA-C*01:02 marked in red). **C** Comparison of HLA-C amino acid allele divergence between Niedertiefenbach individuals with and without the allele HLA-C*01:02 (*$P < 0.05$).

In line with studies investigating the health status of Neolithic populations in central Europe[45], the Niedertiefenbach individuals showed numerous unspecific skeletal lesions that are indicative of physical stress, including malnutrition, and infections[38]. Interestingly, we did not detect any pathogens. This observation is consistent with aDNA-based findings describing only relatively few sporadic cases of infectious diseases for the Neolithic period[46].

The HLA class I and II dataset generated for Niedertiefenbach was relatively small and thus precluded sophisticated statistical analysis. However, relative to contemporary German populations some striking shifts in allele frequencies could be observed (Table 1 and Supplementary Data 3). Interestingly, several of the alleles that are less common today (e.g., A*02:01, B*27:05, C*01:02, DQB1*03:01, and DRB1*08:01) are associated with higher resistance to viral pathogens (e.g., HIV, HCV, influenza A, and herpesvirus)[47–50] and often also with higher susceptibility to bacterial infections or complications thereof[50–52]. When we

traced the most frequent alleles of the class I and class II loci through time (by checking their proxy SNP alleles in published aDNA datasets), it became apparent that five HLA-B, -C, and -DRB1 alleles were a hallmark of HGs, but not of later-dating farmers (Fig. 3a and Supplementary Data 4). Their high frequencies in Niedertiefenbach may thus reflect the considerable HG-related ancestry proportion in the population. The alleles were potentially maintained at this frequency at that time because of their functional uniqueness, including a higher sequence divergence as well as a unique repertoire of presented antigens. Both of these properties should confer an advantage in fighting diverse viruses and other pathogens[53]. Later on, they may have lost their relative fitness advantage, e.g., because pathogens adapted to these most common alleles in a process of negative frequency-dependent selection[54] and were replaced by alleles beneficial against newly emerging human pathogenic bacteria, such as *Yersinia pestis*. For the once common allele HLA-C*01:02, no protective effect against an infectious agent is known

today. Thus, it is tempting to speculate that it had evolved in defense of a pathogen that is no longer virulent or that went extinct as has been described for Neolithic HBV lineages[55].

Another notable difference concerns the HLA allele DRB1*15:01. It is widespread in present-day Europeans (ca. 15%), but absent in Niedertiefenbach samples. This allele predisposes to mycobacterial infections (tuberculosis and leprosy)[56]. In disease studies, the SNP allele rs3135388-T is often used as a marker for DRB1*15:01[57]. In published aDNA datasets[25], rs3135388-T was also found to be absent in all European Paleolithic, Mesolithic, and Neolithic populations analyzed. It seemed to appear for the first time only during the Bronze Age. Since then, its initially high frequency (~20%) has decreased to the present low levels (Supplementary Fig. 10). This finding raises the intriguing possibility that the allele might have been incorporated into the European gene pool as part of the steppe-related ancestry component in the Final Neolithic and Bronze Age. Given the limited size of the ancient sample, these considerations remain speculative and await corroboration, as HLA data from further ancient populations become available.

The advent of farming and subsequent shifts in pathogen exposure are thought to have radically changed the immune genes in early agriculturalists[25]. The immune response of the Niedertiefenbach collective appears geared towards fighting viral agents. To what extent this HLA profile was due to the specific demographic history of the Niedertiefenbach population (i.e., the high HG ancestry proportion) or typical of Neolithic communities in the fourth millennium remains to be clarified. Overall, our study showed that the HLA-repertoire of modern Europeans was established quite recently, sometime during the last 5000 years, and may also have been shaped by population admixture.

By applying a comprehensive genomics approach to individuals interred in the WBC-associated collective burial in Niedertiefenbach, we discovered that the community, which used this site for about 100 years, was genetically heterogeneous and carried both Neolithic and HG ancestry. The mixture of these two components likely occurred at the beginning of the fourth millennium, indicating important demographic and cultural transformations during that time in western Europe. This event may also have affected the immune status of the admixed population and its descendants for generations to come.

## Methods

**Samples**. The archaeological site and osteological and paleopathological results were described previously[16,17,38].

**Osteological analysis**. Estimations for skeletal sex and age at death (Supplementary Data 1) were performed on temporal bones and were based on osteological standards for cranial elements[58,59]. We focused on crania as the human remains were commingled in the collective burial. Age spans were kept wide (e.g., 20–40 years; adult ++) and sex was expressed as tendency (female > male) rather than category as only few diagnostic attributes are present on temporal bones (e.g., cranial sutures for age, mastoid process for sex). Of the 19 studied well-preserved samples, 14 were consistent in both osteological and genetic sex. This level of correlation was good, given the available material.

**Radiocarbon dating**. Collagen was extracted and dated from 25 human bone samples, originally collected for aDNA analysis, and each attributed to a different individual. Dating was performed following standard protocols at the Leibniz Laboratory for AMS Dating and Isotope Research, Kiel (details in Supplementary Information)[15].

**aDNA extraction and sequencing**. Surface contaminants from petrous bones and teeth were removed with bleach solution. Partial uracil-DNA-glycosylase-treated sequencing libraries were prepared from bone or tooth powder-derived DNA extracts following previously established protocols[60]. Sample-specific index combinations were added to the sequencing libraries[61]. Sampling, DNA extraction, and the preparation of sequencing libraries were performed in clean-room facilities of the Ancient DNA Laboratory in Kiel. Negative controls were taken along for the

DNA extraction and library generation steps. The libraries were paired-end sequenced using 2 × 75 cycles on an Illumina HiSeq 4000. Demultiplexing was performed by sorting all the sequences according to their index combinations. Illumina sequencing adapters were removed and paired-end reads were merged if they overlapped by at least 11 bp. Merged reads were filtered for a minimum length of 30 bp.

**Pathogen screening**. All samples were screened with MEGAN[62] and the alignment tool MALT[63] for their metagenomic content[55].

**Mapping and aDNA damage patterns**. Sequences were mapped to the human genome build hg19 (International Human Genome Sequencing Consortium, 2001) using BWA 0.7.12[64] with a reduced mapping stringency parameter "-n 0.01" to account for mismatches in aDNA. Duplicates were removed. C to T misincorporation frequencies were obtained using mapDamage 2.0[65], to assess the authenticity of the aDNA fragments[18]. After the validation of terminal damage, the first two positions from the 5′ and 3′-ends of the fastq-reads were trimmed off.

**Genotyping**. Alleles were drawn at random from each of the 1,233,013 SNP positions[1,2,25] in a pseudo-haploid manner using a custom script[66]. Datasets were filtered for at least 10,000 SNPs to be considered for further analysis[5].

**Genetic sex determination**. Genetic sex was determined based on the ratio of sequences aligning to the X and Y chromosomes compared to the autosomes[67]. Females are expected to have a ratio of 1 on the X chromosome and 0 on the Y chromosome, whereas males are expected to have both X and Y ratios of 0.5. We used an upper threshold of 0.016 of the fraction of reads mapping to the Y chromosome for females and an upper bound of 0.075 for males[68]. A sample was called female when the CI was below 0.016 or it was called male when the CI was above 0.075.

**Contamination estimation and authentication**. Estimation of DNA contamination was performed on the mitochondrial level using the software Schmutzi[69] and in males by applying ANGSD[70], to investigate the amount of heterozygosity on the X chromosome.

**Principal component analysis**. The genotype data of the Niedertiefenbach collective were merged with previously published genotypes of 5519 ancient and modern individuals genotyped on the aforementioned 1,233,013 SNPs using the program *mergeit* from the *EIGENSOFT* package[71]. PCA analysis was performed using the software *smartpca*[71] by projecting the genotype datasets of the Niedertiefenbach and all other ancient individuals on the principal components calculated from the genotype datasets of 59 present-day West-Eurasian populations from the Affymetrix Human Origins dataset. The principal components were calculated using the "lsqproject" option. No shrinkage correction was applied.

**ADMIXTURE analysis**. Prior to ADMIXTURE analysis, we used Plink (v1.90b3.29) to filter out SNPs with insufficient coverage (0.999) and a minor allele frequency below 5%. Linkage disequilibrium pruning was performed to filter out SNPs at an $R^2$ threshold of 0.4 using a window size of 200 and a step size of 25. We ran ADMIXTURE (version 1.3.0)[20] on the same populations as used in the PCA analysis. The number of ancestral components ranged from 4 to 8. Cross-validation was performed for every admixture model.

**Admixture dating**. The source code of ALDER (v1.03)[22] was modified to decrease the minimal number of samples needed for the analysis as described here: https://www.diva-portal.org/smash/get/diva2:945151/FULLTEXT01.pdf. Thus, reference populations with only a single individual could be included. The following reference populations were used for Niedertiefenbach: Anatolia_Neolithic, OrienteC_HG, Croatia_Mesolithic_HG, Bichon, Blatterhohle_MN, Koros_Hungary_EN_HG, Serbia_HG, Serbia_Mesolithic_Neolithic, Narva_LT, Iron_Gates_HG, Loschbour, Iberia_HG, Latvia_EN, Baalberge_MN France_MN, Latvia_HG. These populations were used as closest unadmixed genetic proxies for possible parental sources based on the qpAdm results. To calculate calendar dates of admixture, we multiplied the obtained number of generations with an assumed generation time of 29 years[24]. The applied model does not take into consideration multiples waves, continuous admixture or admixture of populations that were already admixed[3]. Thus, the obtained dates reflect only a minimal number of generations. Furthermore, we ran the software DATES[23] with default parameters (binsize: 0.001; maxdis: 1; seed: 77; runmode: 1; jackknife: YES; qbin: 10; runfit: YES; affit: YES; lovalfit: 0.45), to confirm our results obtained by ALDER.

**F3 outgroup statistics**. F3 outgroup statistics were run as a part of the *Admixtools* package[21] in the form of $f_3$(*Niedertiefenbach; test, Mbuti*) using for *test* the same populations as in the PCA and ADMIXTURE analyses.

**qpAdm analysis**. qpAdm analysis was run on transition-filtered genotypes that were previously prepared for ADMIXTURE analysis as described above. We ran 48 different combination models of Niedertiefenbach as a 2-way admixture, as 3-way admixture models appeared to be less feasible, indicating that the third component was excessive. The following populations were used as outgroups: Mbuti, Ust' Ishim, Kostenki14, Mal'ta (MA1), Han, Papuan, Onge, Chukchi and Karitiana and —optionally—Villabruna, Croatia_Mesolithic_HG and OrienteC_HG (Supplementary Data 2). We then ran qpAdm for each individual using the following HGs as proxies for the HG component to see how its amount varied among the individuals: Koros_Hungary_EN_HG; Bichon; Serbia_HG; Iron_Gates_HG; Iberia_HG; WHG; Blätterhöhle; Loschbour (Supplementary Fig. 9).

**Kinship analysis**. Kin relatedness was assessed using READ[34] and lcMLkin[72]. READ identifies relatives based on the proportion of non-matching alleles. lcMLkin infers individual kinship from calculated genotype likelihoods. A pair of individuals was regarded related only if evidence of relatedness was independently provided by both programs (Supplementary Fig. 8).

**Determination of mitochondrial and Y chromosome haplotypes**. Sequencing reads were mapped to the human mitochondrial genome sequence rCRS[73]. Consensus sequences were generated in Geneious (v. 9.1.3) using a default threshold of 85% identity among the covered positions and a minimum coverage of 3. HAPLOFIND[74] was applied to assess mitochondrial haplotypes from the consensus sequences and yHaplo[75], to determine Y chromosome haplotypes in male individuals using the ISOGG 2016 standard.

**Calling of phenotypic SNPs**. We generated a pile-up of reads mapping to the positions of the selected phenotypic SNPs with samtools mpileup (v. 1.3) to see how many reads supported a particular allele in each individual.

**HLA typing and analysis**. We used a previously established HLA capture and HLA-typing pipeline[29]. In addition, we applied OptiType[30] for automated HLA class I and II typing. For pairs of related individuals (first- and second-degree relatives), we then removed one of the individuals based on the maximum number of reads supporting the HLA call in either of the two individuals, to obtain an HLA dataset with only unrelated individuals. Samples with low coverage of the HLA region were also excluded. Only alleles that were consistently called by both methods were considered for the analysis. For comparing the ancient HLA allele pool with a representative modern allele pool, we used a cohort of 3219 healthy German individuals and imputed HLA genotypes at second field level of resolution from high-density SNP data following an established procedure[76]. We also calculated allele divergence for HLA-B and HLA-C genotypes using the Grantham distance matrix and the publicly available tool GranthamDist[32]. The Grantham score incorporates physicochemical properties of the different amino acids and was shown to be the most suitable proxy for functional divergence in peptide binding among HLA class I variants[32]. HLA-B and HLA-C allele-specific binding of viral peptides (Supplementary Table 2) was predicted with NetMHCpan v4[77] using the proposed affinity rank threshold of 2%. For allele pool comparison of HLA-B and HLA-C, we included only alleles with frequency higher than 0.0217, the detection limit for the data of Niedertiefenbach with $N = 23$ individuals. Allele pool composition was compared for each locus separately using an ANOSIM (from the vegan package)[78] with 1000 permutations run in R v3.4.2[79]. The phylogenetic tree of amino acid sequences from alleles at HLA-B and HLA-C was calculated in MEGA X[80] using the maximum-likelihood method with JTT substitution matrix and default settings.

**Statistics and reproducibility**. Statistical analyses are based on data generated as part of this study, specifically aDNA sequence data of individuals from Niedertiefenbach, as well as data from publicly available previous studies, as indicated in each specific subsection of methods and results. Statistical tests were chosen depending on the specific question, data structure, and data distribution, generally relying on more conservative non-parametric tests. Sample sizes are reported where applicable, usually referring to the number of individuals in a given group. Individuals represent biological replicates. For a subset of the historical specimens, aDNA was extracted from both petrous bone and tooth as technical replicates, which yielded the same results. Technical procedures of sample handling and molecular processing, as well as parameter settings in computational processing and analysis of sequence and genotype data are reported in each subsection and are based on previously published methods.

**Reporting summary**. Further information on research design is available in the Nature Research Reporting Summary linked to this article.

## Data availability
The aligned sequences are available through the European Nucleotide Archive under accession number ERP118364. The analyzed skeletal material belongs to the Landesamt für Denkmalpflege Hessen, hessenARCHÄOLOGIE.

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

## Acknowledgements

This study was funded by the Deutsche Forschungsgemeinschaft (DFG, German Research Foundation) through the CRC 1266 (project number 2901391021), grant LE 2593/3-1 to T.L.L. and Germany's Excellence Strategy—EXC2167–390884018. A.N. was supported by the Dorothea Erxleben Female Investigator Award of the DFG Cluster of Excellence *Inflammation at Interfaces* (EXC306). We thank I. Mathieson for discussion of the results and comments on the manuscript. We are grateful to S. Schiffels, Th. C. Lamnidis, and A. Mittnik for sharing their expertise in ALDER, qpAdm, and kinship analyses, and their advice on result verification. J.B. and F.P. were funded by the International Max Planck Research School for Evolutionary Biology.

## Author contributions

B.K.-K., A.N., and C.R. developed the idea for this study. K.F. analyzed the human skeletal remains. L.B. and B.K.-K. generated ancient DNA data. A.I., J.S., and B.K.-K. analyzed the

ancient DNA data. A.I., A.S., F.P., L.B., J.D., J.B., D.E., J.C.K., R.B., O.K., T.L.L., A.F., J.K., and B.K.-K. analyzed modern and ancient HLA data. S.S.-L., S.S., J. Meadows, J. Müller, C.R., C.D., M.F., A.I., T.L.L., A.N., and B.K.-K. interpreted the findings. A.I., A.N., T.L.L., and B.K.-K. wrote the manuscript with input from all other authors.

## Funding

## Competing interests
The authors declare no competing interests.
