## [Peer Review File · Communications Biology]

Reviewers' comments:

Reviewer #1 (Remarks to the Author):

In this work, 42 new ancient DNA sequences from the Late Neolithic Wartberg culture from a single site (Niedertiefenbach, present-day Germany) are presented. The authors use several standard and up-to-date ancient DNA analysis techniques and also analyze HLA genotypes (which is not standard in the aDNA field).

Overall, the work is technically competent and the analysis is sound, but I have a number of minor comments to improve analysis and presentation (see below). Overall, I find the manuscript easy to follow and to be well written.

I believe this work is highly relevant and novel, and the data will provide a major contribution for Archaeology and ancient DNA. As a key finding, the authors identify not only an exceptionally high HG component for a late Neolithic population, but also that HG ancestry was quite variable within this population. As the authors describe, this "cline" implies that the admixture is very recent or maybe even ongoing. This finding is extremely intriguing, given the very late date (more than 2000 years after early Central European LBK farming cultures). This new data and finding provides much fuel for the "resurgence of HG" ancestry, and implies that somewhere in or near Central Europe there must have been unknown "refugia" of this quite different HG ancestry. So far, only very few "odd" outliers (e.g. from Blätterhöhle) with Late Neolithic high HG ancestry are known, and to my knowledge this is the first time this Late Neolithic "resurgence" is described for a site with clear archaeological context.

On the other hand, I do not find the HLA analysis very convincing, and to be sometimes highly speculative (especially the parts about selection). In particular, comparing to present-day populations, after several major demographic transitions (such as the appearance of the Steppe ancestry, which was a major population turnover in Europe after 5000 BP), is not very conclusive, as it is hard to disentangle selection and demographic processes such as migrations. Analyzing new data from a single site does not seem to be the ideal place for such an analysis of HLA markers (42 individuals is not a big sample size, even for common alleles), and this analysis would be better done as a meta-analysis. However, I am not an expert on HLA, and if the authors insist on keeping it (since they did the work already) I don't see a scientific reason for cutting it other than that it is not very conclusive.

Overall, the authors both draw upon genetic as well as archeological evidence, which is refreshing to see, and I applaud the authors for that effort.

I have a number of comments that the authors hopefully find helpful to improve the presentation (see below). But I do not identify major "make or break" roadblocks. I first give the "major" comments (1) followed by more minor, technical (but hopefully useful) comments (2-3)

#####

1 Major comments:

1.1 It would help to at least mention absence of Yamnaya type Steppe ancestry. The timing of the site (just before 5000 BP) is starting to overlap the Yamnaya horizon, and falls only few hundred years before this ancestry makes an absolutely major impact in Central Europe. Mentioning that you find 2-way models (with Steppe ancestry as outgroup ideally) and the absence of R1b Y haplogroups are already sufficient evidence. So you do not need to run further analysis, but it is good to mention it explicitly for the general reader.

1.2 You present 25 new radio carbon dates. This is highly useful data by itself, and could be mentioned in the main text (it is somewhat hidden). It seems there were no surprises and everything lines up, but again, this is an important finding by itself.

1.3 You use "Alder", but the up-to-date and more potent follow-up "DATES" package (<https://github.com/priyamoorejani/DATES>) is already widely used. In particular, this follow-up

software is better suited to application for ancient populations as target. If there is sufficient time, I would recommend using this software as it provides improved power and accuracy.

1.4

As HG and Neolithic Ancestry is very diverged, which gives one a lot of power for qpAdm analysis of 2-way models for each individual (with more than 100k SNPs, say). I would strongly recommend running such a model for every individual, and showing a barplot with estimated admixture fractions (and error bars), as often seen in aDNA. These "varying" admixture fractions (as seen on the PCA) are a key part of the paper, and it would really help assessing them with a tool developed for that usecase (qpAdm) and visualizing them.

1.5

Supp. Table 2:

It would really help if you report the raw and calibrated radio-carbon dates in this table, as well as contamination estimates where available. Having this in this machine-readable table will largely help others to use this data, and only increase your impact.

#####

2 Main text comments:

2.1 Abstract L50: "exhibited a distinct human leukocyte antigen gene pool, resulting in an immune response that was primarily geared towards defending against viral infections."

That seems quite speculative, do we really understand HLA that well? For my taste and given the weak evidence that is too definitive. In I295 you use "appears", maybe add a similar quantifier.

2.2 I122-123: State qpAdm p-Values (in brackets) so the expert reader can assess the fit himself.

2.3

I178:

"Nine of the 25 males carried the same Y chromosome haplotype"

Mention which one in brackets. Overall I2c1a1 seems quite interesting by itself, but I do not want to burden you with additional analysis (one can always do more).

2.3

Do you want to discuss the male bias in sex determination? (26/37 males is statistically significant). At least a sentence of speculation why that is could be interesting.

2.4

I313: "where described previously" instead of "elsewhere" has a more positive ring.

2.5

I157-159:

"We observed that five of them, i.e. alleles at HLA-B, -C and -DRB1, appeared at much higher frequencies in HG ($\geq 47\%$) than in the Niedertiefenbach samples or in early farmers (Fig.8)"

Is this statistically significant?

2.6

Two major papers about French (and some German) Neolithic sites were published just last week (<https://advances.sciencemag.org/content/6/22/eaaz5344>, and

<https://www.pnas.org/content/early/2020/05/20/1918034117>). You do not have to co-analyze this data given the advanced state of the manuscript and your local focus, but adding a sentence in the discussion would be useful (especially regarding the resurgence of HG ancestry).

#####

3 Method Comments:

3.1 Genetic sex determination: State the cutoff for genetic sex determination in the methods

3.2

Principal component analysis: State whether shrinkage correction was used (if not it is no problem, but mention it)

And it was genotypes from present-day populations used for creating the PCs, right? And was this the Human Origin dataset? Clarify in methods, otherwise it is not reproducible.

3.3

Y Haplogroup: What ISOGG standard is used to report the Y haplogroups? And requiring 3 reads seem like a very stringent threshold in ancient DNA (but I will not criticize you for being conservative).

3.4

qpAdm: Report what outgroups were used. This is critical information for reproducibility and interpretation, but I could not find this information.

3.5

Table 1: Please give confidence intervals (e.g. 95% confidence intervals) for the frequencies. They are likely large, but that's why it is worth showing them for small sample sizes!

3.6

Admixture Plots are very hard to read: You could increase label size. You do not use the admixture results in any key result, so you could also relegate it to the supplement altogether (where the figure requirements are less stringent).

Reviewer #2 (Remarks to the Author):

Immel et al. present new ancient genome-wide data for 42 individuals from a collective burial located in Niedertiefenbach, Germany, and associated with the Wartberg culture (WBC, 3,500-2,800 BCE). The analyses demonstrate that this early farming community carried a surprisingly large component (40%) of hunter-gatherer (HG) ancestry. Admixture dating indicates that the HG component was most likely introduced during the cultural transformation that led to the WBC around 3,800-3,500 BCE. Comparing allele frequencies at six different HLA loci associated with regulating immune response in humans, the team found significant differences between the Niedertiefenbach individuals and modern Germans, suggesting that the Niedertiefenbach individuals may have been better "equipped" at defending against viral infections. Using proxy SNPs, the team also found that five of the HLA alleles appeared at much higher frequencies in HGs, suggesting that their presence in the Niedertiefenbach individuals may reflect the considerable HG-related ancestry proportion in that population. Immel et al. go on to suggest that this combination of alleles may have lost their relative fitness advantage, for example because pathogens adapted to these most common alleles in a process of negative frequency-dependent selection, or because they were replaced by alleles that proved to be beneficial to ward against newly emerging bacterial pathogens, such as *Y. pestis*.

This is a well-written paper that presents a set of very interesting results. To my knowledge Immel et al. present the first genome-wide data for individuals associated with the Wartberg culture and the finding that this group carried a substantial amount of HG ancestry is certainly worthy of note and should be of interest to the wider ancient DNA/archaeological community. Although their sample size was relatively small, their finding of differences in allele frequencies at HLA loci between the Niedertiefenbach individuals and modern Germans is also interesting because it

suggests that the HLA repertoire of modern Europeans was established relatively recently, ie. some time during the last 5000 years, and that may also have been shaped by population admixture.

In summary, I think that this is a very interesting paper and I am happy recommend it for publication.

Minor comments:

1) In line 94 of the manuscript you refer to Supplementary Table 1, but since you are referring to the data, I assume you mean Supplementary Table 2. I assume it's just a typo but this should be corrected.

2) Since you used shotgun sequencing to generate the data, I recommend that you include more information in Supplementary Table 2. At the very least I would include the number of raw and uniquely mapped reads you generated per sample. This is useful information and should be reported.

3) Since you did not generate whole genomes I suggest you consider rephrasing the title of the manuscript to reflect that. I don't think it will take anything away from the study, but it is more accurate.

Point-by-point response

We would like to thank the two reviewers for taking the time to review our manuscript.

Reviewer #1:

On the other hand, I do not find the HLA analysis very convincing, and to be sometimes highly speculative (especially the parts about selection). In particular, comparing to present-day populations, after several major demographic transitions (such as the appearance of the Steppe ancestry, which was a major population turnover in Europe after 5000 BP), is not very conclusive, as it is hard to disentangle selection and demographic processes such as migrations. Analyzing new data from a single site does not seem to be the ideal place for such an analysis of HLA markers (42 individuals is not a big sample size, even for common alleles), and this analysis would be better done as a meta-analysis. However, I am not an expert on HLA, and if the authors insist on keeping it (since they did the work already) I don't see a scientific reason for cutting it other than that it is not very conclusive.

We understand the hesitation of the reviewer towards the interpretation of the HLA data, given its limited sample size. On the other hand, this is the first ever population data of HLA for such an ancient period, and we therefore believe that it is of significant interest, at the very least to spark further research. However, we have now toned down the interpretation further.

1 Major comments:

1.1 It would help to at least mention absence of Yamnaya type Steppe ancestry. The timing of the site (just before 5000 BP) is starting to overlap the Yamnaya horizon, and falls only few hundred years before this ancestry makes an absolutely major impact in Central Europe. Mentioning that you find 2-way models (with Steppe ancestry as outgroup ideally) and the absence of R1b Y haplogroups are already sufficient evidence. So you do not need to run further analysis, but it is good to mention it explicitly for the general reader.

We have inserted two sentences stating the absence of steppe ancestry in the Niedertiefenbach population: *“The radiocarbon dates (3,300-3,200 cal. BCE) place the site only a few hundred years before the arrival of the steppe ancestry in central Europe. Interestingly, we do not observe any genetic evidence for an admixture with steppe components (e.g. no feasible two-way models with steppe ancestry or the absence of the Y haplogroup R1b).”* (page 10).

1.2 You present 25 new radio carbon dates. This is highly useful data by itself, and could be mentioned in the main text (it is somewhat hidden). It seems there were no surprises and everything lines up, but again, this is an important finding by itself.

We have inserted two sentences about the 25 new radiocarbon dates at the beginning of the Results section (page 5). However, all the information and the dating model are described in more detail in an article accepted by *Radiocarbon* (Ref 15: Meadows et al. 2020).

1.3 You use "Alder", but the up-to-date and more potent follow-up "DATES" package (<https://github.com/priyamoorejani/DATES>) is already widely used. In particular, this follow-up software is better suited to application for ancient populations as target. If there is sufficient time, I would recommend using this software as it provides improved power and accuracy.

We have followed the reviewer's suggestion and have used DATES. We used default parameters as now also described in the Material and Methods section. Indeed, we obtain a similar number of generations (16 +/- 2) for the proposed admixture of Loschbour HG and Anatolian Neolithic farmers, re-affirming our previous results from ALDER. We added the result to the Supplementary Information. Further, we have added the following sentence to the main text (page 6): *"We confirmed our result using the software DATES which yielded a similar number of generations (16.6 +/- 2.65) (Supplementary Information)."*

1.4 As HG and Neolithic Ancestry is very diverged, which gives one a lot of power for qpAdm analysis of 2-way models for each individual (with more than 100k SNPs, say). I would strongly recommend running such a model for every individual, and showing a barplot with estimated admixture fractions (and error bars), as often seen in aDNA. These "varying" admixture fractions (as seen on the PCA) are a key part of the paper, and it would really help assessing them with a tool developed for that usecase (qpAdm) and visualizing them.

We have followed the reviewer's recommendation and have included the qpAdm 2-way models for each individual in Supplementary Fig. 9. We have mentioned the "varying" admixture fractions in the population in the Discussion on page 11: *"The WBC-associated population in Niedertiefenbach represents a genetically diverse group with a very broad range of HG proportions (as seen in Fig. 2 and in the qpAdm 2-way models for each individual in Supplementary Fig. 9)."*

1.5 Supp. Table 2: It would really help if you report the raw and calibrated radio-carbon dates in this table, as well as contamination estimates where available. Having this in this machine-readable table will largely help others to use this data, and only increase your impact.

The requested information (i.e. raw radiocarbon dates and the data needed to estimate a possible contamination, e.g. lab ID, collagen yield, delta ¹³C, delta ¹⁵N) is shown in Supplementary Table 1. The calibrated radiocarbon dates are presented in Supplementary Fig. 2. In addition, all the information and the dating model are described in detail in an accepted article (Ref 15: Meadows et al. 2020).

2 Main text comments:

2.1 Abstract L50: "exhibited a distinct human leukocyte antigen gene pool, resulting in an immune response that was primarily geared towards defending against viral infections." That seems quite speculative, do we really understand HLA that well? For my taste and given the weak evidence that is too definitive. in l295 you use "appears", maybe add a similar quantifier.

We have toned down the message in the Abstract. It now reads as follows (page 2): *"The Niedertiefenbach individuals exhibited a distinct human leukocyte antigen gene pool, possibly reflecting an immune response that was geared towards detecting viral infections."*

2.2 I122-123: State qpAdm p-Values (in brackets) so the expert reader can assess the fit himself.

We have inserted the p values on page 6, line 122-124.

2.3 I178: "Nine of the 25 males carried the same Y chromosome haplotype" Mention which one in brackets. Overall I2c1a1 seems quite interesting by itself, but I do not want to burden you with additional analysis (one can always do more).

We have added the Y haplotype I2c1a1 in brackets (page 8, I178).

2.3 Do you want to discuss the male bias in sex determination? (26/37 males is statistically significant). At least a sentence of speculation why that is could be interesting.

Indeed, our genetic sex distribution in the sample indicated a considerable excess of males (70%) amongst adults and subadults, which has not been described for other Neolithic populations. Since we followed a random sampling strategy, such an excess is noteworthy and may reflect a burial bias. We have added these sentences to the Discussion (page 12).

2.4 I313: "where described previously" instead of "elsewhere" has a more positive ring.

We have changed the sentence as recommended.

2.5 I157-159: "We observed that five of them, i.e. alleles at HLA-B, -C and -DRB1, appeared at much higher frequencies in HG ($\geq 47\%$) than in the Niedertiefenbach samples or in early farmers (Fig.8)" Is this statistically significant?

Five out of seven is not statistically significant ($P=0.25$), but we have now tested statistical significance for the change in allele frequency between the Neolithic samples and modern Germans for all these 12 common alleles. Most frequency changes are significant after multiple-testing correction. This new information is now included in Table 1.

2.6 Two major papers about French (and some German) Neolithic sites were published just last week (<https://advances.sciencemag.org/content/6/22/eaaz5344>, and <https://www.pnas.org/content/early/2020/05/20/1918034117>). You do not have to co-analyze this data given the advanced state of the manuscript and your local focus, but adding a sentence in the discussion would be useful (especially regarding the resurgence of HG ancestry).

We refer to these two new publications when discussing the resurgence of the HG ancestry.

3 Method Comments:

3.1 Genetic sex determination: State the cutoff for genetic sex determination in the methods

We added the following sentence to the corresponding section (page 18): *“We used an upper threshold of 0.016 of the fraction of reads mapping to the Y chromosome for females and an upper bound of 0.075 for males⁶⁸. A sample was called female when the confidence interval (CI) was below 0.016 or it was called male when the CI was above 0.075.”*

3.2 Principal component analysis: State whether shrinkage correction was used (if not it is no problem, but mention it) And it was genotypes from present-day populations used for creating the PCs, right? And was this the Human Origin dataset? Clarify in methods, otherwise it is not reproducible.

We thank the reviewer for pointing this out. We have inserted three sentences as follows (pages 18-19): *“Principal component analysis (PCA) was performed using the software smartpca by projecting the genotype datasets of the Niedertiefenbach and all other ancient individuals on the principal components calculated from the genotype datasets of 59 present-day West-Eurasian populations from the Affymetrix Human Origins dataset. The principal components were calculated using the 'lsqproject' option. No shrinkage correction was applied.”*

3.3 Y Haplogroup: What ISOGG standard is used to report the Y haplogroups? And requiring 3 reads seem like a very stringent threshold in ancient DNA (but I will not criticize you for being conservative).

We have used the default parameters of yHaplo which is the ISOGG standard 2016.

3.4 qpAdm: Report what outgroups were used. This is critical information for reproducibility and interpretation, but I could not find this information.

We have mentioned the outgroups in the Methods section about qpAdm (page 20): *“The following populations were used as outgroups: Mbuti, Ust` Ishim, Kostenki14, Mal'ta (MA1), Han, Papuan, Onge, Chukchi and Karitiana.”*

3.5 Table 1: Please give confidence intervals (e.g. 95% confidence intervals) for the frequencies. They are likely large, but that's why it is worth showing them for small sample sizes!

We have added a column to Table 1 with the 95% confidence intervals for the HLA allele frequencies of the Niedertiefenbach individuals.

3.6 Admixture Plots are very hard to read: You could increase label size. You do not use the admixture results in any key result, so you could also relegate it to the supplement altogether (where the figure requirements are less stringent).

We have deleted Fig. 3 as recommended. All the admixture results are shown in Supplementary Fig. 3.

Reviewer #2:

Minor comments:

1) In line 94 of the manuscript you refer to Supplementary Table 1, but since you are referring to the data, I assume you mean Supplementary Table 2. I assume it's just a typo but this should be corrected.

Thank you for pointing out this oversight. We have corrected the sentence accordingly (now line 93 in the manuscript).

2) Since you used shotgun sequencing to generate the data, I recommend that you include more information in Supplementary Table 2. At the very least I would include the number of raw and uniquely mapped reads you generated per sample. This is useful information and should be reported.

We have included the number of raw and uniquely mapped human reads as well as the damage patterns in Supplementary Table 2.

3) Since you did not generate whole genomes I suggest you consider rephrasing the title of the manuscript to reflect that. I don't think it will take anything away from the study, but it is more accurate.

We have changed the title to "Ancient DNA study reveals a distinct HLA allele pool and population transformation in Neolithic Europe".

REVIEWERS' COMMENTS:

Reviewer #1 (Remarks to the Author):

The authors have effectively and competently answered my comments, adding new analysis and updating the descriptions where necessary. I believe the manuscript is a strong unit of publication now, and I want to applaud the authors for presenting ancient DNA evidence within a solid archeological framework.

The extraordinary high (and variable) HG admixture continues to be a striking result which really adds to the picture of interaction between Hunter Gatherer and Early Farmer ancestry. I am very optimistic that this pattern will meet wide-spread interest, so I cannot wait to see this work being officially published.

I only have a number of very minor comments left (some of which I just noticed in this read, for which I apologize). Some of them are purely "stylistic" (which effectively are none of my business as a reviewer). If the authors think these comments strengthen the manuscript, they can include them - if not it is of course fine as is.

Scientific Comments:

A1) Relatedness detection: Do I read that right and you actually only find the triplet and some really distant relationships? You only mention the triplet in L181. However the lack of any 2nd degree and only one 3rd degree relatives in a collective burial and the majority of individuals not being related to anyone is ...highly intriguing. Such a pattern is indicative of a large population. Importantly, it means that the grave is not linked to close kinship (which is observed in some other Neolithic burials). Also, was the "full sibling" triplet found in close physical proximity? The labels which are very "close" would suggest so.

A2) Given this lack of relatedness and the male bias - was there any sign of violence on the skulls? ("perimortem" injuries?) I assume not, but in case you have the info it would be helpful to mention this lack of evidence when you describe the lesions in L259.

A3) Y chromosomes: Please add the "ISOGG 2016" you mention in the rebuttal to the Y description in the methods ("Determination of mitochondrial and Y chromosome haplotypes"). The exact labeling of Y changes between ISOGGs, which is typically source of much confusion (e.g. in the current up-to-date ISOGG 2019-2020 I2c1a1 changed to I2a2a1a - which can be confusing if you don't mention which standard you are using. It's absolutely okay to use an older one and much work does).

A4) Also double check the "9" Y chromosomes (L178). It seems that some of the Y haplogroup calls are very high level and could not be refined (e.g. B/T, C/F and F are all ancestral to I and the individuals likely just don't have sufficient reads from any of the sub-lineages to make any more refined calls), but that does not mean these males are necessarily different than the I2 calls. Also, the single A sounds a bit unbelievable (it's Sub-Saharan Africa only at present-day and in aDNA record) and most likely contamination/low coverage. Overall the data seems entirely consistent with being made up of a few (I think four) I2 lineages. And fewer Y haplogroups is even stronger support for a "patrilineage".

A5) Table 1: The addition of Confidence Intervals is a major improvement, that really makes the differences seem striking (where they are). The CIs for modern data will be minuscule, but it would be good (and consistent) to explicitly show that too and add one column. Overall, I cannot wait to see a overall meta-study of HLA patterns, and whether changes are mainly explained by migration and population turnover or selection (there is much sequence data out there).

A6) The added individuals qpAdm figs in the Supp. Info are striking. Do you plot 1 SE error bars? Mention this in the figure caption. And it seems that adding Villabruna as right pop reduces the uncertainty substantially based on Supp. Tab. 3, did you do that here? Generally, adding a few more relevant early HG right pops could help to reduce the SE.

A7) Supp. Tab. 3: The pattern that when adding Villabruna as outgroup you can "destroy" the Iberia HG model but the Loschbourg/Blaetterhoehle admixture p-Value going up is strong evidence for NW-European HG ancestry being the source (which makes perfect sense of course but is nice to see). And since you mention Villabruna - it cannot hurt to add the other right pops too in that Supp. Tab. 3 (there is plenty of free real estate).

A8)

Quite a few citations are in German. While this is highly relevant context, it could help the international readership of this journal if for the general statements citations are paired up with a English publication (e.g. L63).

Stylistic Comments:

B1) The title "Ancient DNA study reveals a distinct HLA allele pool and population transformation in Neolithic Europe" does not refer to the Wartberg Culture and does not give away "genome-wide" nor the overall population genetic aspect of the study.

B2) The abstract could mention that the HG ancestry is also highly variable among individuals in the collective burial (30-60%) and not only the mean, which I personally find a striking result: The collective grave wasn't "homogeneous".

B3) Fig 1 and Fig 2 could be merged into one panel. In Figure 2, the PC2 is flipped, most ancient DNA Western Eurasian PCAs have Steppe being "up". The "sign" of the PCA is pretty much non-informative and is essentially random. So flipping this axis would not change anything but make the PCA more comparable to others. Moreover, usually a West Eurasian PCA is plotted as a "square" and not a wide rectangle. Experienced readers would really benefit from "standardizing" this figure. And now it's really stylistic: You could add an "ellipse" around Wartberg and label it "Wartberg Culture" in your PCA, which would serve as a visual clue to highlight the target of your study.

Reviewer #2 (Remarks to the Author):

Thank you for addressing our comments. I think the manuscript reads really well. It's an exciting study and I look forward to seeing it in print!

Minor edits:

Line 77 and line 88 (and throughout the rest of the ms e.g. lines 203 and 205) - I would always use the present tense when reporting results.

Line 146: I would change "lactose intolerance" to "lactase non-persistence"

Line 231 - I would specify admixture dating analysis here and refer to the SI.

Line 232 - Delete "an".

Line 236 - I suppose HG should be plural here, so HGs?

Line 239 - Data is plural, so should be followed by a plural verb as at the start of the next paragraph (The genetic data ... shed light...).

Line 249 - Again HGs instead of HG?

Line 261 - Maybe "detect" is better than "observe"?

Line 266 - Here "observed" is stronger than "seen".

Line 273 - Again HGs instead of HG?

Line 298 - ... as HLA data ... become available.

Line 304 - Delete "was".

Line 312 - Delete "components".

Final revisions for manuscript COMMSBIO-20-1118A

Reviewer #1 (Remarks to the Author):

The authors have effectively and competently answered my comments, adding new analysis and updating the descriptions where necessary. I believe the manuscript is a strong unit of publication now, and I want to applaud the authors for presenting ancient DNA evidence within a solid archeological framework.

The extraordinary high (and variable) HG admixture continues to be a striking result which really adds to the picture of interaction between Hunter Gatherer and Early Farmer ancestry. I am very optimistic that this pattern will meet wide-spread interest, so I cannot wait to see this work being officially published.

I only have a number of very minor comments left (some of which I just noticed in this read, for which I apologize). Some of them are purely "stylistic" (which effectively are none of my business as a reviewer). If the authors think these comments strengthen the manuscript, they can include them - if not it is of course fine as is.

Scientific Comments:

A1) Relatedness detection: Do I read that right and you actually only find the triplet and some really distant relationships? You only mention the triplet in L181. However the lack of any 2nd degree and only one 3rd degree relatives in a collective burial and the majority of individuals not being related to anyone is ...highly intriguing. Such a pattern is indicative of a large population. Importantly, it means that the grave is not linked to close kinship (which is observed in some other Neolithic burials). Also, was the "full sibling" triplet found in close physical proximity? The labels which are very "close" would suggest so.

Reply to the reviewer:

Based on our results, the collective burial primarily contained the remains of unrelated individuals and does not appear to be linked to close kinship. We have mentioned this aspect in the Discussion (L252-254): "Overall, the genomic data indicate that the gallery grave was mainly used by not closely related people who may have lived in various neighboring locations. This observation is supported by the large number of mtDNA haplogroups."

In fact, there is only one case of three first-degree relatives in the sample. The IDs of the three individuals (KH150620, KH150622 and KH150623) are lab numbers and do not reflect any proximity in the collective grave. As described in the Methods section, the human remains were very much commingled in the collective burial. Therefore, even proximity in the archeological context would not necessarily indicate that the dead were buried next to each other or at the same time.

A2) Given this lack of relatedness and the male bias - was there any sign of violence on the skulls? ("perimortem" injuries?) I assume not, but in case you have the info it would be helpful to mention this lack of evidence when you describe the lesions in L259.

Reply to the reviewer:

As mentioned above, the remains in the collective burial were not only commingled, but also very often poorly preserved, thus precluding any diagnosis of "perimortem" injuries.

A3) Y chromosomes: Please add the "ISOGG 2016" you mention in the rebuttal to the Y description in the methods ("Determination of mitochondrial and Y chromosome haplotypes"). The exact labeling of Y changes between ISOGGs, which is typically source of much confusion (e.g. in the current up-to-date ISOGG 2019-2020 I2c1a1 changed to I2a2a1a - which can be confusing if you don't mention which standard you are using. It's absolutely okay to use an older one and much work does).

Reply to the reviewer:

We have added the ISOGG standard to the paragraph "Determination of mitochondrial and Y chromosome haplotypes" in the Methods section (L444): "HAPLOFIND⁷⁴ was applied to assess mitochondrial haplotypes from the consensus sequences and yHaplo⁷⁵ to determine Y chromosome haplotypes in male individuals using the ISOGG 2016 standard."

A4) Also double check the "9" Y chromosomes (L178). It seems that some of the Y haplogroup calls are very high level and could not be refined (e.g. B/T, C/F and F are all ancestral to I and the individuals likely just don't have sufficient reads from any of the sub-lineages to make any more refined calls), but that does not mean these males are necessarily different than the I2 calls. Also, the single A sounds a bit unbelievable (it's Subsahran Africa only at present-day and in aDNA record) and most likely contamination/low coverage. Overall the data seems entirely consistent with being made up of a few (I think four) I2 lineages. And fewer Y haplogroups is even stronger support for a "patrilineage".

Reply to the reviewer:

Thank you for pointing out this error. We absolutely agree that the high-level Y haplogroup calls look dubious and are likely due to the lack of coverage. We have therefore replaced all unrefined (high-level) calls (A, BT, CF, CT, F, I) by "NA" in our Supplementary Table 2 and also changed the main text as follows (L177): "We noted 29 different mitochondrial DNA (mtDNA) haplogroups and 5 Y chromosome haplotypes, all of which belonged to haplogroup I2 (now Supplementary Data 1). Ten of the 16 males for whom high-resolution Y haplotype information could be generated carried the same haplotype (I2c1a1)."

A5) Table 1: The addition of Confidence Intervals is a major improvement, that really makes the differences seem striking (where they are). The CIs for modern data will be minuscule, but it would be good (and consistent) to explicitly show that too and add one column. Overall, I cannot wait to see a overall meta-study of HLA patterns, and whether changes are mainly explained by migration and population turnover or selection (there is much sequence data out there).

Reply to the reviewer:

The CIs for the modern German population have been added to Table 1.

A6) The added individuals qpAdm figs in the Supp. Info are striking. Do you plot 1 SE error bars? Mention this in the figure caption. And it seems that adding Villabruna as right pop reduces the uncertainty substantially based on Supp. Tab. 3, did you do that here? Generally, adding a few more relevant early HG right pops could help to reduce the SE.

Reply to the reviewer:

Initially, we did not use Villabruna in the individuals for the qpADM plots, but following the reviewer's suggestion, we have now added Villabruna to the rightpops. This has indeed decreased the standard error bars. Adding more early HG groups such as Mesolithic HG from Croatia and Late Upper Paleolithic HG from Sicily (Oriente C), however, has led to increased error bars. Therefore, we do not show the latter results. We have updated the individuals' plots and the figure caption after including Villabruna (now Supplementary Data 2 and Supplementary Fig. 9).

A7) Supp. Tab. 3: The pattern that when adding Villabruna as outgroup you can "destroy" the Iberia HG model but the Loschbourg/Blaetterhoehle admixture p-Value going up is strong evidence for NW-European HG ancestry being the source (which makes perfect sense of course but is nice to see). And since you mention Villabruna - it cannot hurt to add the other right pops too in that Supp. Tab. 3 (there is plenty of free real estate).

Reply to the reviewer:

Following the reviewer's suggestion, we have run the qpADM models including Villabruna, Croatia_Mesolithic_HG and OrienteC_HG in the rightpops (see reply to the previous comment). We have obtained more feasible models. We have updated Supplementary Data 2 accordingly.

A8) Quite a few citations are in German. While this is highly relevant context, it could help the international readership of this journal if for the general statements citations are paired up with a English publication (e.g. L63).

Reply to the reviewer:

The WBC is basically a German archaeological phenomenon (s. its distribution in Figure 1). Therefore, all the relevant literature is also in German.

Stylistic Comments:

B1) The title "Ancient DNA study reveals a distinct HLA allele pool and population transformation in Neolithic Europe" does not refer to the Wartberg Culture and does not give away "genome-wide" nor the overall population genetic aspect of the study.

Reply to the reviewer:

Following the reviewer's and editor's suggestion we have changed the title to "**Genome-wide study of a Neolithic Wartberg grave community reveals distinct HLA variation and hunter-gatherer ancestry.**"

B2) The abstract could mention that the HG ancestry is also highly variable among individuals in the collective burial (30-60%) and not only the mean, which I personally find a striking result: The collective grave wasn't "homogeneous".

Reply to the reviewer:

We have replaced the words "on average 40%" with "(34-58%)" in the Abstract.

B3) Fig 1 and Fig 2 could be merged into one panel. In Figure 2, the PC2 is flipped, most ancient DNA Western Eurasian PCAs have Steppe being "up". The "sign" of the PCA is pretty much non-informative and is essentially random. So flipping this axis would not change anything but make the PCA more comparable to others. Moreover, usually a West Eurasian PCA is plotted as a "square" and not a wide rectangle. Experienced readers would really benefit from "standardizing"

this figure. And now it's really stylistic: You could add an "ellipse" around Wartberg and label it "Wartberg Culture" in your PCA, which would serve as a visual clue to highlight the target of your study.

Reply to the reviewer:

We have changed Figure 2 as recommended (flipped PC2, removed the label, fitted the PCA into a square and highlighted the Wartberg Culture with an ellipse). We have decided not to merge Figure 1 and Figure 2 into one figure, otherwise the labels of the PCA would become illegible.

Reviewer #2 (Remarks to the Author):

Thank you for addressing our comments. I think the manuscript reads really well. It's an exciting study and I look forward to seeing it in print!

Minor edits:

Line 77 and line 88 (and throughout the rest of the ms e.g. lines 203 and 205) – I would always use the present tense when reporting results.

Reply to the reviewer:

We would like to adhere to our style of using past tense throughout because we want to make the story behind our research obvious. For story telling, past tense is the means of choice. Additionally, to stay within the story and with one tense, makes it easier for the reader to follow.

Line 146: I would change "lactose intolerance" to "lactase non-persistence"

Reply to the reviewer:

We have changed "lactose intolerance" to "lactase non-persistence".

Line 231 - I would specify admixture dating analysis here and refer to the SI.

Reply to the reviewer:

We have included this information and the sentence (L230-L231) now reads as follows: This scenario is tentatively supported by the **admixture** dating analysis (**Supplementary Information**).

Line 232 - Delete "an".

Reply to the reviewer:

We have deleted the indefinite article as suggested.

Line 236 - I suppose HG should be plural here, so Hgs?

Reply to the reviewer:

Throughout the manuscript, we have changed HG to HGs when the abbreviation is used in plural.

Line 239 - Data is plural, so should be followed by a plural verb as at the start of the next paragraph (The genetic data ... shed light...).

Reply to the reviewer:

Throughout the manuscript, we have now used data consistently as a plural noun.

Line 249 - Again HGs instead of HG?

Line 261 - Maybe "detect" is better than "observe"?

Line 266 - Here "observed" is stronger than "seen".

Line 273 - Again HGs instead of HG?

Line 298 - ... as HLA data ... become available.

Line 304 - Delete "was".

Line 312 - Delete "components".

Reply to the reviewer:

We have followed all the above suggestions and have made the respective changes in the text.